# Anthocyanins Formulated with Carboxymethyl Starch for Gastric and Intestinal Delivery

**DOI:** 10.3390/molecules27217271

**Published:** 2022-10-26

**Authors:** Luiz Bruno De Sousa Sabino, Francesco Copes, Solène Saulais, Edy Sousa De Brito, Ivanildo José Da Silva Júnior, Tien Canh Le, Mircea Alexandru Mateescu, Diego Mantovani

**Affiliations:** 1Department of Chemical Engineering, Federal University of Ceará, Av. Mister Hull, 2977, Campus Universitário do Pici, Fortaleza 60356-000, Ceará, Brazil; 2Laboratory for Biomaterials and Bioengineering, Department of Mining, Metallurgy and Materials Engineering & CHU de Québec, Regenerative Medicine Division, Laval University, Quebec City, QC G1V 0A6, Canada; 3Department of Chemistry, Université du Québec à Montréal, CP 8888, Succ. Centre-Ville, Montreal, QC H3C 3P8, Canada; 4Embrapa Agroindústria Tropical, R. Dra Sara Mesquita, 2270, Fortaleza 60511-110, Ceará, Brazil

**Keywords:** anthocyanins, carboxymethyl starch, bioactive agent delivery, monolithic tablets, *Syzygium cumini*

## Abstract

Anthocyanins obtained from jambolan have been used as active agents in different carboxymethyl starch-based tablet formulations and their release profiles evaluated in simulated gastric fluids (SGF) and simulated intestinal (SIF) fluids. Structural analysis highlighted a strong interaction between anthocyanins and carboxymethyl starch, evidenced by scanning electron microscopy and infrared analysis. Tablet dissolution behavior varied according to the pH of the media, being controlled by the swelling and/or erosion of the polymeric matrix. Various formulations for immediate, fast, and sustained release of anthocyanins for 30 min, 2 h and 12 h of dissolution have been developed. It was found that monolithic carboxymethyl starch tablets loaded with powdered jambolan extract efficiently afforded the complete delivery (100% of anthocyanins) to different sites of the simulated gastrointestinal tract and ensured the stability of these pigments, which maintained their antioxidant activity.

## 1. Introduction

Anthocyanins (ACNs) are water-soluble pigments, widely distributed in plants, and responsible for the blue, purple, red, and pink shades of flowers, fruits, and leaves. These molecules present interesting biological properties and are particularly known known as powerful natural antioxidants [1,2]. In fact, in vivo and in vitro tests have shown that ACNs extracts exert antioxidant, anti-inflammatory, anti-tumoral and anti-atherosclerotic activities [2]. These pigments may improve endogenous antioxidant defences by enhancing the activities of superoxide dismutase (SOD) and glutathione peroxidase (GPx) antioxidant enzymes, and indirectly, with the decrease of DNA oxidative damage and the formation of endogenous ROS through the inhibition of NADPH oxidase and xanthine oxidase [2]. These properties are attributed to the natural electron deficiency of ACNs, resulting in an elevated reactivity and ability to act as a proton donor, stabilizing reactive oxygen species (ROS) [3,4]. Due to their unquestionable benefits to human health, the inclusion of sources of ACNs in a regular diet is highly recommended [5]. ACNs are non-essential nutrients, so their average daily intake is not well established. In the United States, it has been estimated that a typical diet contains a range of 12.5 to 215 mg of ACNs/day. The ACNs can be obtained from the ingestion of fruits such as blueberries (3.87–4.87 mg ACN s/g), blackberries (2.45 mg ACN s/g), raspberries (1.16 mg ACN s/g) and strawberries (0.35 mg ACN s/g) and their derivatives, or wine (0.72 mg ACN s/mL [5,6]. Together with these well-known sources of ACNs, in this research, the jambolan (*Syzygium cumini* L.), a tropical and underexploited fruit, was used as a source for the extraction of ACNs.

Jambolan is a fruit widely distributed in Latin America that presents an attractive purple colour and an exotic flavour, both provided by the combination of organic acids, sugar, and phenolic compounds [7]. Some studies indicated that jambolan extracts present several pharmacological gastroprotective, anticancer and antidiabetic properties, attributed to their high content of phenolic agents, especially ACNs. In fact, previous research demonstrated that, depending on the cultivar, the concentration of ACNs of jambolan can vary from 28.5 to 131.84 mg/100 g, which is higher than that found in other sources [8,9]. Despite being an easily obtainable raw material and presenting a composition rich in phytochemicals, the exploitation of jambolan is limited in relation to other tropical fruits. Thus, the development of studies that use this fruit as a potential source for the extraction of biocompounds is increasingly important [8,9,10,11].

The effectiveness of ACNs depends on their bioavailability, which corresponds to the amount of ACNs that will reach the systemic circulation, usually referring to the absorption and metabolism of these compounds by the organism [12,13,14,15]. Intrinsic characteristics of the gastrointestinal tract (GIT), e.g., physiological temperature [12] and variation of pH, can cause rearrangements in the ACN structures, affecting their bioavailability and, consequently, their bioactivity [3,7,14,15]. In response to pH variation in the GIT, five species of ACNs can co-exist in pH-dependent equilibrium (Figure 1). In an aqueous solution, the red flavylium cation is the most stable ACN form, being predominant in the acid conditions of the stomach (pH 1–3). Once reaching the basic environment of the intestine, the flavylium cation changes to colourless carbinol pseudo-base and, in smaller amounts, to yellow chalcone and blue quinoidal forms (Figure 1) [8,9]. Thus, during the passage through the GIT, ACNs are most likely to exist in different forms, which tend to affect their bioavailability [6].

In humans, the complete mechanism of ACN absorption and metabolism are still unclear. Experimental evidence suggested that ACNs are mainly absorbed in the upper gastrointestinal tract (stomach and the initial part of the small intestine) [12,17]. The ACNs show a rapid diffusion (around 20–25%) into gastric mucosa, which would be facilitated by their binding with the organic anion carrier bilitranslocase [18]. Thus, the pigments advance to the systemic circulation, in their native or metabolized forms, being available to exert their biological activities. The ACN fractions that are not absorbed in the stomach reach the small intestine (up to 65%), where they are absorbed, efficiently metabolized (to their glucuronidated, sulfated, or methylated derivatives), reaching the circulation or excreted into bile or urine [19]. In general, the overall bioactivity of the ACNs results from the synergistic effect of their native structure or their degradation products and metabolites, which eventually reach the systemic circulation. If ACN residence time in these organs is not enough for metabolization, the undigested fraction will be eliminated in the gut lumen and degraded due to the higher intestinal pH and microbiota action. The extensive degradation of ACN native structures may limit the amount of the parent glycoside available to exert the overall bioactivity [20,21].

To guarantee the bioactivity of ACNs, a practical strategy would be to increase their retention time in sites of the GIT where they are actively absorbed [20]. In this light, the development of ACN delivery systems may represent a viable solution to increase the retention of these molecules for an adequate time at the desired GIT location and, therefore, enhance their bioavailability and activity [21]. Several ACN delivery systems were proposed in the literature, including the lipid-based [22], protein-based [23] and polysaccharide-based [24] anthocyanin complexes. Between these systems, the polysaccharide-ACN complexes are the most promising due to the notable stabilization conferred to ACNs, thus avoiding their extensive degradation in different biological environments [25].

Different polysaccharides have been studied with the aim of protecting and boosting ACNs’ functionality [26,27]. ACNs are hydrophilic molecules and are compatible with water-based gel formulations with gums, maltodextrin, and starches, for example. Starch is a natural, biodegradable, and cheap polymer, which, however, presents a limited industrial application due to its poor processability, structural stability and mechanical properties in its native form [28,29]. Nonetheless, starch properties can be modulated by physical, enzymatic, and chemical modifications, improving its use for the formulation of different products such as tablets [30] and microspheres [29], for example. Carboxymethyl starch (CMS) is an anionic polymer obtained from the introduction of carboxylic groups in starch structure by an etherification reaction with monochloroacetic acid [28]. The CMS has been proposed as a novel pH-sensitive excipient [31]. Previous studies showed that in simulated gastric acidity, CMS outer layers are compacted (due to protonation of carboxylic groups) providing gastro-resistant formulations. Then, in intestinal fluids, CMS-based materials are deprotonated and facilitate the release of loaded active compounds, due to the related swelling of the matrix [28,31]. Despite a lack of literature regarding ACNs:CMS complexation, the ionically stabilized matrix resulting from that interaction could control the release of ACNs according to the environmental pH, ensuring their stability/availability in different zones of the GIT.

The notable potential that ACNs represent for human health demands the development of strategies to enhance their stability and bioavailability, including preliminary studies that may offer basic information about the behaviour of ACN delivery systems. In this research, ACNs and CMS were associated to produce monolithic tablets for delivery of ACNs into gastrointestinal simulated media. The release of ACNs was evaluated and formulations were developed for their controlled release at different sites of the gastrointestinal tract. To the best of our knowledge, this is the first study to evaluate the use of CMS as an excipient for the delivery of ACNs.

## 2. Results and Discussion

### 2.1. Characterization of the Samples

#### 2.1.1. Scanning Electron Microscopy (SEM)

The electron micrographs of native starch, CMS and the ACNs:CMS complex are presented in Figure 1. Starch presented a granular morphology (Figure 1(A1,A2)) characterized by irregular sizes, round shapes, and smooth surfaces. Differently, CMS micrographs (Figure 1(B1,B2)) displayed irregularly shaped granules and pronounced agglomerations of smaller particles. CMS wrinkled surface may be the result of a structural reorganization promoted by the interaction carboxyl-carboxyl groups (dimerization) and between carboxylic and hydroxyl groups [30].

The morphologies obtained from the ACNs and CMS association are shown in Figure 1(C1,C2). The SEM images of the powders obtained by freeze-drying suggested that the morphology of ACNs:CMS differs from that of native starch and of CMS. The SEM micrographs presented irregular structures with different sizes, which may be attributed to freeze-drying processes, grinding, and sieving to obtain the powder. Moreover, the pores present on the material surface can be the result of air bubbles collapsing during the drying process [26]. The more compact structure of the ACNs:CMS complex suggests a strong association between both components which probably will enhance the stability of the ACNs.

#### 2.1.2. FTIR Analysis

The FTIR spectra and the description of identified bands are presented in Figure 2. For native starch and carboxymethyl starch (CMS), new characteristic bands, ascribed to the symmetrical (1417 cm^−1^) and asymmetrical (1600 cm^−1^) stretching vibration of carboxylate groups (-COO^−^), are proofs of carboxymethylation of the native starch (Figure 2A).

For the evaluation of CMS matrix behaviour in the dissolution media, tablets have been analysed (Figure 2A): after 2 h incubation in SGF, and after 2 h in SGF (pH = 1.2) followed by 6 h in SIF (pH = 6.8). Following the SGF incubation, bands at 1600 and 1417 cm^−1^ decreased as a consequence of the protonation of carboxylate to carboxylic acid, showing a new absorption band at 1724 cm^−1^ (Figure 2A). After subsequent incubation in SIF, the signal at 1724 cm^−1^ disappeared following the deprotonation of carboxylic acid groups to carboxylate and a recurrence of bands at 1592 and 1417 cm^−1^ was also observed, in agreement with previous data on deprotonation of carboxylic groups [31,32].

The ACNs characteristic spectrum (Figure 2B) showed bands related to their chemical structure: at 3303 cm^−1^ characteristics of the -OH groups, from 692.3 to 916.9 cm^−1^ related to the vibration of aromatic groups, and a specific band at 2934 cm^−1^ for the C-H stretching of the benzene ring. The absorption detected at 1024 cm^−1^ was identified with the aromatic ring C-H deformation. Bands at 1670 and 1455 cm^−1^ were assigned, respectively, to the benzopyran aromatic ring carbonyl group (C=O) vibration and to the stretching vibration (C=C) of the aromatic ring. The band at 1230 cm^−1^ was assigned to the pyran rings stretching, a flavonoid compounds signature, while the signal at 1340 cm^−1^ was assigned to C-O angular deformations of phenols [33,34].

Regarding the ACNs-CMS complexes, signals nearby 1650 cm^−1^ (F1: 1698 cm^−1^; F2: 1692 cm^−1^; F3 cm^−1^: 1646 cm^−1^; F4: 1708.6 cm^−1^) were the result of the interaction of -OH and C=O from ACNs with COO^−^ groups from CMS (Figure 2B). ACNs’ typical peak at 1230 cm^−1^ disappeared, while the aromatic ring signal linked to the C-H deformation (at 1024 cm^−1^) changed in all spectra, especially on F4, where its intensity increased at 1012 cm^−1^. The changes observed in F4 signals may be due to the freeze-drying process that enhanced the interaction between ACNs and CMS. The evident changes observed in all the tablet spectra between 1000–1800 cm^−1^ were ascribed to the ortho alteration of ACNs’ aromatic rings and may indicate the association between the components of the tablets [35,36]. In Figure 2 we propose a hypothetical mechanism for a better understanding of ACNs:CMS association. Supported by the SEM results found in this research, it was suggested that hydroxyl and/or carboxylate groups present in the CMS matrix interact with ACNs through hydrogen bonds and electrostatic interactions resulting in a stable ACNs:CMS complex.

The infrared spectra of mannitol and PVP as well as more explanations of their association with ACNs extract are presented in Appendix A as Appendix A [39,40].

Therefore, these results, and in particular the strong changes observed in the ACNs’ structure following the association with CMS, confirm the interactions between the components of the monolithic tablets.

#### 2.1.3. Erosion and Fluid Uptake

To characterize erosion and fluid uptake in the formulated tablets, unloaded CMS tablets behaviour was evaluated in both SGF and SIF fluids to predict the ACNs’ release. The CMS tablets were incubated for 2 h in SGF (pH = 1.2) or in SIF (pH = 6.8). Figure 3 shows how the CMS fluid uptake and erosion was higher in SIF (266% and 11.3%, respectively) than in SGF medium (131.3% and 7.03%, respectively). The percentage of residual CMS in the matrix provides indications of the polymer amount dissolved in the fluids. After incubation in SGF or in SIF, tablets presented 93% and 88% of their remaining mass, respectively.

The differences between the fluid uptake and erosion of the tablets in SGF or in SIF (Figure 3), are attributed to the different pH of these fluids. The high acidity of SGF contributes to the protonation of outer layers of CMS carboxylic groups and, consequently, to the stabilization of the matrix through hydrogen bonds. This induces a compaction of the tablet, resulting in lower fluid uptake and tablet erosion. Differently, in SIF the CMS deprotonation results in ionisation of the carboxylic -COO^−^ groups, resulting in higher fluid intake and tablet erosion [31,32]. These results are corroborated by the FTIR spectra presented in Figure 2A.

### 2.2. In Vitro Anthocyanins Release Study

#### 2.2.1. ACNs’ Release from the CMS-Based Tablets in Simulated Gastric and Intestinal Media

The ACNs’ release from F1, F2, F3 and F4 tablets was evaluated in SGF (pH = 1.2), and in SGF followed by SIF (pH = 6.8) to simulate the gastric and intestinal residence (Figure 4). The detailed composition of the tablets F1, F2, F3 and F4 can be found in Appendix A.

Generally, the dissolution profiles showed how ACN release was influenced by the pH of the medium. The release of ACNs was followed in SGF up to 9 h (Figure 4A) in order to better understand the behaviour of ACNs:CMS associations and to provide preliminary data for eventual gastro-retentive CMS microspheres as carriers [20,29]. Results showed a sustained release in SGF, with a burst effect (about 40%) for the first 2 h. Then the release rate decreased after 5 h and remained stable at 7 h of dissolution. This is probably due to the higher compaction of the outer layer of the protonated tablets in gastric acidity, thus limiting ACN release.

The amounts of the ACNs released in SGF medium were 81.6%, 84.2%, 80.2% and 85.0% for the F1, F2, F3 and F4 tablets, respectively. Considering that the ACNs’ concentration did not decrease during all the incubation time, these results are in agreement with previous reports that indicated the structural stability of ACNs under SGF conditions [20,41]. This stabilization of ACNs by ACNs-CMS interactions, in SGF, was also indicated by the remaining coloration of the tablets due to an amount of unreleased ACNs (about 18%) after 9 h of dissolution (Figure 5). Moreover, the tablets maintained their shape in SGF and did not swell. The faster release (burst effect) in the first 60 min in SGF was found also with other formulations of ACNs such as various microcapsules of ACNs with a higher ACNs release of 47.7% after 2 h of incubation in SGF [24,42,43]. The lower burst effect in our formulations can be mainly attributed to the protonation of the CMS matrix and the ACNs:CMS interaction.

To mimic the transit from the stomach to the intestine, tablets have been incubated in SGF for 2 h followed by an incubation in SIF medium until complete dissolution (approximately 9 h). The dissolution profiles at 37 °C are shown in Figure 4B.

In general, the dissolution data suggested that ACNs are progressively released from the tablets in different proportions as a function of the pH of the medium. The first 2 h of dissolution in SGF (pH = 1.2) presented an ACN release in the range of 35% to 45%. Then, the tablets were transferred into SIF (pH = 6.8) and the ACNs’ liberation was continued following a progressive release pattern until the end of the essay. Similar dissolution behaviours were reported by Oidtmann et al. [44]. The authors evaluated three systems for the encapsulation of bilberry ACNs (*Vaccinium myrtillus* L.) and reported an increase in the release of these pigments of 64%, 77% and 85% after their transfer from SGF to SIF. This fact can be attributed to pH which tunes the interaction between ACNs and CMS. Due to the SIF neutral pH, the overall positive charges of ACNs are decreased as well as their ionic binding with CMS, which allows the effective dissolution of ACNs in this medium.

Mannitol and PVP are excipients commonly used for drug processing, acting in the protection of the bioactive agents, as well as enhancing their release through the formation of pores within the dosage form [39,40]. With our formulations with CMS, it was found that mannitol did not present a remarkable influence on the ACNs’ release considering that the amount released from F2 was the lowest (61.2% at 9 h) among the tablets. The F3, supplemented with PVP, released 81.3% of its ACNs content, just behind F4 which released more than 90% of ACNs in the gastrointestinal simulated media at the end of dissolution. A previous study evaluated the release of ACNs from chitosan microcapsules in GIT simulated media and a release of 30.6% was found after 6 h of dissolution, which was lower than the value presented by F2 at the same dissolution time (47.3%) [24].

In this research, no pronounced degradation of ACNs was verified during SGF/SIF dissolution, as suggested by the progressive quantification of ACNs in their flavylium form, over time. According to Betz et al. [45], it is desirable to deliver ACNs to the intestine in their most stable flavylium cation form and to reduce thereby their degradation which is more extensive when these pigments reach this organ in their quinoidal or carbinol parental forms. In contrast, Oidtmann et al. [44] reported the degradation of 42% of the total ACNs released after 1 h 30 min of dissolution in the gastric or intestinal medium.

Considering the data obtained from ACN release profiles, F4 was chosen as the most suitable formulation for ACNs’ delivery over a programmable time. Accordingly, the rest of the study was continued with excipients with this formulation.

#### 2.2.2. Anthocyanins Release Modeling

The kinetic parameters of ACNs’ release for the dissolution in SGF (2 h) followed by the dissolution in SIF (SGF/SIF) fitted to the Korsmeyer–Peppas model and are shown in Table 1.

Values of *k*, *n*, and the correlation coefficient (R^2^), were obtained by plotting the log (Mt/M_∞_) against log(t). The data were used to determine the ACNs’ release mechanism for the model with Mt/M_∞_ ≤ 70%. It is known that the drug release from matrices such as CMS (i.e., swellable polymers) is complex and not completely clarified. The release can happen due to either pure diffusion or controlled erosion, but most often a combination of the two phenomena may occur. In our study, due to a correlation coefficient (R^2^) higher than or equal to 0.90, all the data showed a good fit with the Korsmeyer–Peppas equation, therefore implying a combined effect of diffusion and erosion mechanisms [46,47].

A high correlation coefficient (R^2^ ≥ 0.95) was noted for the results obtained in SGF. *n* values in the range of 0.50–0.90, corresponding to an anomalous (non-Fickian) diffusion mechanism, were observed. This was probably due to the protonation and compaction of the tablets. The liberation was faster in SGF and with release patterns close to each other for all formulations. These results are in agreement with the data reported by Prasad and Shelar [48]. Celli et al. [20] developed an alginate-based gelling system to evaluate ACN release in the gastric medium. Different results were found in this research, where the ACN release mechanism in SGF was controlled by Fickian diffusion. The difference between these results can be attributed to the different swollen states of CMS and alginate in the acid environment, which affect directly the ACN release mechanism.

Regarding the SGF/SIF dissolution, the *n* values ranged from 0.41 to 0.80, like those obtained for the release in SGF. F2 and F4 samples presented *n* values close to 0.45, hinting at a Fickian diffusion with a diffusion-controlled release for both types of tablets. The *n* values of F1 and F3 tablets, conversely, suggested an anomalous (non-Fickian) diffusion release. However, in F1 (*n* = 0.50) the diffusion seems to be the main release mechanism, whereas for F3 (*n* = 0.80) it is erosion. Finally, the dissolution data in SGF/SIF for all the tested formulations fit the Korsmeyer–Peppas model (R^2^ ≥ 0.95). Moreover, all these results in various dissolution media support the fact that the release of ACNs from the CMS tablets is pH sensitive.

#### 2.2.3. Optimization of Formulation

The F4 tablets formulation was optimized to develop a programmable ACN delivery system in the gastric or intestinal medium.

Initially, the formulations F4.1, F4.2, F4.3, F4.4 and F4.5 were derived from the F4 formulation. These formulations contained different proportions of disintegrating agents (Appendix A) to mediate the total release of ACNs in SGF in 30 min. As shown in Figure 6, the F4.5 tablets, containing Explotab^®^ and microcrystalline cellulose (MCC), were totally disintegrated in 30 min, releasing 100% of their ACNs content (Figure 6). The other experimental condition tested did not exhibit the same pattern and released less than 50% of the ACNs after 30 min (Appendix A). The fitted proportion of both disintegrating agents induced a fast hydration of the polymeric matrix, allowing a total disintegration, and, consequently, a complete release of ACNs in F4.5 tablet. Explotab^®^ is a commercial low substituted cross-linked carboxymethyl starch that can promote a rapid fluid uptake, resulting in increased granule volume and, therefore, tablet disintegration [49].

To prolong the ACNs’ release in SGF, formulations F4.6 and F4.7 were developed from F4, supplemented with microcrystalline cellulose (MCC) and croscarmellose sodium (CCS), as described in Appendix A. The tablets were kept in SGF for 2 h and the amount of ACNs released in the dissolution medium was measured every 30 min. From the dissolution profile presented in Figure 6, it is possible to see that both samples started to release the ACNs after 1 h and, at 2 h of dissolution, 100% of the ACNs were released in the medium. The complete release of ACNs was reached due to the synergistic effect of MCC and CCS used in the formulation. The use of MCC can result in a high hydrophilicity and wetting of the tablets, due to an increase in their porosity (erosion) while CCS, a cross-linked polymer of carboxymethylcellulose, promotes an efficient disintegration by the rapid wicking and swelling and upon contact with water [50,51].

ACNs are commonly stable in the stomach, presenting a fast absorption via the bili-translocase carrier [18]. Different studies indicate that the peak of plasma ACN concentration is achieved within 30 min to 2 h after ingestion [52]. Therefore, the development of F4.5 (immediate release), F4.6 and, F4.7 (release up 2 h) tablets may be a promising strategy to prolong the gastric retention of ACNs, while maintaining their natural structural form, resulting in better absorption and bioavailability.

For the sustained release of ACNs in the intestinal tract, formulation F4.8 was developed, based on tablet F4, and supplemented with adequate proportions of MCC and CCS (Appendix A). The ACNs release was evaluated in SGF (for 2 h) and then in SIF (until the complete disintegration of the tablets) with measurements done every 30 min in SGF and every 60 min in SIF. A progressive release of ACNs was observed from the first 30 min until 12 h, when 100% of the ACNs were released in SIF medium (Figure 7). Thus, the dissolution pattern from F4.8 tablets suggested a sustained intestinal role of 100% of ACNs in 12 h of dissolution. The proper sustained release was reached by the combination of the erosion and swelling due to the fitted proportion of MCC, and CCS, which led to the progressive release of ACNs during the tablet’s gastric and intestinal residence.

The ACNs that are not absorbed in the stomach, upon reaching the small intestine, are absorbed, biotransformed by the enterocytes, transported to the hepatocytes, metabolized, and then distributed to the organism [10,39]. The remaining fraction is degraded in the colon by the action of local microbiota and then absorbed [13]. The F4.8 tablet can be an interesting ACN delivery system to afford sustained release of the ACNs, increasing their bioavailability/bioactivity over time in the intestine. The sustained release could prevent the saturation of the transporters involved in ACN uptake, increasing their absorption, and maintaining their bioactivities in vivo [43,53]. In addition, sustained release of ACNs in the intestine could decrease the irreversible formation of carbinol species (promoted by the long exposure of ACNs to neutral pH) which cannot be rearranged to the flavylium cation form, thus reducing the bioactivity of these molecules.

### 2.3. Antioxidant Capacity of the ACNs Released in Simulated Media

The ability of the released ACNs from F4 tablets to scavenge the free radical 2,2-azino-bis (3-ethylbenzothiazoline-6-sulfonic acid) diammonium salt (ABTS^•+^) was evaluated in SGF/SIF media (Figure 7). As expected, the increase of ACN concentration in the dissolution medium resulted in a decreased free-radical concentration. The antioxidant activity of ACNs is related to their easy oxidation, which may induce the deactivation and stabilization of reactive ABTS^•+^ radicals by the donation of a free electron or hydrogen atoms [16,54].

The ACNs are molecules known for their low structural stability to neutral-alkaline pH. In this context, the antioxidant activity of ACNs released from CMS-tablets during 8 h of dissolution (2 h in SGF/6 h in SIF) clearly indicated the CMS stabilizing effects, when compared with the antioxidant activity of the free ACNs pure extract (with a concentration equivalent to F4). After 8 h of SGF/SIF dissolution (Figure 7), an antioxidant activity of 327 µmoles Trolox.mL^−1^ was observed for F4 (ACNs formulated with CMS) compared to 72.5 µmoles Trolox.mL^−1^ for the free-ACNs extract.

As previously discussed, the long exposure of ACNs to the intestinal medium may promote an irreversible structural rearrangement from flavylium cation to carbinol, resulting in the loss of their bioactivity. Thus, the continuous controlled release of ACNs into the intestinal tract may allow a sufficient concentration of these compounds at their absorption sites and prevent their irreversible degradation at the neutral intestinal pH (7–7.5).

## 3. Materials and Methods

### 3.1. Chemicals

The ABTS^•+^ (2,2-azino-bis (3-ethylbenzothiazoline-6-sulfonic acid) diammonium salt (purity 98%), Trolox (6-hydroxy-2,5,7,8-tetramethyl-chroman-2-carboxylic acid), 97% absolute ethanol, D-mannitol, trifluoracetic acid (TFA, 99%), microcrystalline cellulose (MCC), Arabic gum, and sodium chloroacetate (SCA, 98%) were obtained from Sigma Aldrich (Saint Louis, MO, USA). High amylose corn starch (Hylon VII) was kindly provided by Ingredion (Westchester, IN, USA). Croscarmellose sodium (Solutab^®^, CMC, 98%) was purchased from Blanver (São Paulo, SP, Brazil). Explotab^®^ and Prosolv^®^ were obtained from JRS pharma (Patterson, NY, USA) and polyvinylpyrrolidone (PVP) from International Specialty Products (Wayne, WA, USA). The other chemicals were reagent grade.

### 3.2. Plant Material and Anthocyanins (ACNs) Extraction

Jambolan fruits were obtained from a domestic orchard localized in Cascavel, Ceará, Brazil. Seeds were removed and the edible portion (30% of fruit) underwent a freeze-drying process in a Labconco Freeze Dry-5 dryer (Kansas, MO, USA) at −50 °C under pressure 5 mtorr (9.67 × 10^−5^ psi). Then, from the obtained material a powder was obtained and stored at −5 °C until use.

A Ultronique QR500 probe ultrasound extractor (Indaiatuba, SP, Brazil) coupled with a 13 mm titanium tip was used to extract the ACNs. Aqueous ethanol solution (80:20 *v*/*v*, ethanol:water) acidified with 0.1% of trifluoracetic acid was used as solvent accordingly to Sabino et al. [55]. Plant material and solvent were mixed in a 250 mL jacketed beaker (ratio fruit: solvent was 1:15 *w*/*v*) and subjected to ultrasound. The temperature was controlled with a thermostatic water bath (30 °C ± 0.1) through the beaker jacket. The ultrasound power (300 W) and extraction time (7.5 min) were controlled. Finally, the extract was filtered, and the solvent removed under reduced pressure in a Buchi R-215 rotavapor (New Castel, DE, USA) at 40 °C.

### 3.3. Synthesis of Carboxymethyl Starch (CMS)

Carboxymethyl starch was prepared (Figure 3) as described by Assaad and Mateescu [31] with minor changes. 150 g of high amylose corn starch (Hylon VII) was mixed for hydration in 50 mL of pure water in a beaker (2 L) under continuous stirring (Servodyne Mixer, 50000-40, IL, USA) at 50 °C. Then, 500 mL of 2.0 M NaOH were added maintained the stirring (30 min) for starch gelatinization. Carboxymethylation occurred by adding 15 g of sodium chloroacetate in a minimal volume of water. After 1 h, 550 mL of distilled water were added to lower the temperature and the reaction was stopped by adding 120 mL of acetic acid (up to pH = 3.0).

The synthesized CMS was precipitated using aqueous methanol (80% *v*/*v*) and washed repeatedly with 2 L of methanol (80%) up to a conductivity of 50 μS/cm or lower. The CMS was then washed with 1 L of pure methanol and afterwards with 1 L of pure acetone on a Büchner funnel and air-dried at 35 °C for 24 h. The powder was then sieved on a 300 µm screen and stored in plastic bottles kept free of humidity.

### 3.4. Tablet Preparation

For the tablet formulations, half of the extract of ACNs was freeze-dried while the remaining half was maintained in liquid form. Both samples were stored at −5 °C until further use.

#### 3.4.1. Preparation of Anthocyanins Powders

Beyond CMS, the main excipient, mannitol and polyvinylpyrrolidone (PVP) were added to the formulations to evaluate their effects on ACN release. For tableting by dry compression, the ACN powder was directly mixed with CMS for F1 formulation. Alternatively, the liquid extract of the ACNs was mixed with mannitol or PVP, which were fillers for the F2 and F3 formulations, respectively.

For the ACNs-Mannitol (ACNs:Man) preparation (F2), 60 mL of aqueous mannitol (20% *w*/*v*) was gently homogenized with 40 mL of the concentrated extract of ACNs. The ACNs:PVP was obtained similarly (F3) but using a PVP aqueous solution (20% *w*/*v*). The ACN liquid extract was mixed with CMS powder in the proportion of 2:1 (% *v*/*w*) to obtain the formulation F4. The ACNs:Man, ACNs:PVP and ACNs:CMS mixtures were kept under constant stirring until completely homogeneous. The mixtures were frozen (−80 °C), freeze-dried, crushed, and sieved on a 300 µm screen. The samples were stored in a dark glass bottle, kept at −20 °C and free from humidity and light.

#### 3.4.2. Formulations

The four formulations F1, F2, F3 and F4 were prepared according to the compositions presented in Appendix A. F1 tablets were obtained by direct compression of ACNs and CMS dried powders in equal proportion. For F2 and F3 tablets, CMS was mixed with ACNs:Man or ACNs:PVP anthocyanin powders, respectively, in the same proportions. The F4 tablets were produced using the ACNs:CMS powder described in Section 3.4.1.

The dissolution profiles showed that the F4 tablet was the most suitable to control the ACNs release; consequently, this formulation was retained and optimized for further dissolution studies. For this purpose, the combination of different disintegrant agents was evaluated (Appendix A) for new formulations. The formulations F4.1, F4.2, F4.3, F4.4 and F4.5 were developed for immediate gastric release (30 min in SGF), F4.6, F4.7 for fast gastric release (2 h in SGF) and F4.8 was conceived for the sustained release in the GIT (2 h in SGF and then 10 h in SIF).

#### 3.4.3. Tablet Preparation

Tablets of 500 mg were prepared with ACNs and CMS only or in association with mannitol or PVP (Figure 8). Before tabletting, the components of each formulation were thoroughly mixed and sieved (300 µm screen). The tablets (12 mm diameter) were obtained by direct compression on a Carver hydraulic press (Wabash, IN, USA) using flat-faced punches at a pressure of 2.5 tonnes.

### 3.5. Anthocyanins Dissolution Assay

Dissolution tests were performed in SGF (pH = 1.2) and in SIF (pH = 6.8) prepared following USP method [56]. In vitro dissolution tests were carried out in a G25 Controlled Environment Incubator Shaker (Alt, Boston, USA) at 37 °C under agitation at 100 rpm. F1, F2, F3 and F4 tablets dissolution was followed: *(i)* in SGF up to ACNs’ complete release, and (*ii*) in SGF for 2 h and then in SIF up to ACNs’ complete release. 2 mL aliquots were collected every 30 min and the absorbance was measured at 280 nm (Lambda 750 Spectrophotometer, Parkin-Elmer, Massachusetts, USA). SIF or SGF media were used as control accordingly. The ACNs concentration in the collected media was determined by plotting against using calibration curves prepared in SGF (R^2^ = 0.9982) and SIF (R^2^ = 0.9973). Purified ACN extract at known concentrations, ranging from 0.125 to 4 mg/mL, was used as standard.

The data obtained from ACNs’ release kinetics were fitted to the well-known Korsmeyer–Peppas Equation (1), frequently used to describe the release behaviour from polymeric systems when the mechanism is not well known or when more than one type of release phenomenon is involved [57].
(1)$$\frac{M_{t}}{M_{\infty }}={\mathrm{kt}}^{n}$$
where M_t_ is the amount of released drug at time t and M_∞_ is the total amount of loaded drug, and n is the diffusion exponent that corresponds to the transport mechanism. M_t_/M_∞_ corresponds to the fraction of anthocyanins released at the time t; k is a kinetic constant related to the properties of the matrix, the properties of the anthocyanins, and the geometric characteristics of the dosage form and n is the release exponent characteristic of the anthocyanins release. The value of n was used to evaluate the ACN release mechanism. For instance, the value of n = 0.45 indicates a diffusion-controlled release (Fickian diffusion) while n = 0.89 corresponds to a swelling-controlled release (case-II transport). Values between 0.45 and 0.89 refer to anomalous diffusion or non-Fickian diffusion, corresponding to the combination of both phenomena [46].

### 3.6. Characterization of the Tablets

#### 3.6.1. Scanning Electron Microscopy (SEM)

The morphology of native starch (Hylon VII), CMS, and ACNs:CMS complex (F4) was evaluated using a Hitachi S-3400N electron microscope (Oxford instruments, High Wycombe, UK) with secondary electron (SE) and backscattered electron (BSE) detectors for observation.

#### 3.6.2. Fourier Transform Infrared Spectroscopy (FTIR)

Fourier Transform Infrared (FTIR) spectroscopy was performed using a Nicolet 4700 (Madison, WI, USA). Spectra were recorded from 4000 to 400 cm^−1^ at a 2 cm^−1^ resolution and with a total of 32 scans.

#### 3.6.3. Erosion and Fluid Uptake

Erosion and fluid uptake of the excipient were evaluated. Briefly, CMS tablets (500 mg) were prepared and incubated in the same conditions as described for the dissolution tests. Tablets were incubated for 2 h in SGF or in SIF.

Then, tablets were removed from the medium, wiped with paper to eliminate the water excess, and weighed before freeze-drying (wet tablets) and after freeze-drying (dried tablets). Erosion and fluid uptake were expressed in percentages and calculated by Equations (2) and (3) as described by Calinescu et al. [28].
(2)$$Erosion\left(\%\right)=\frac{W_{i}-W_{d}}{W_{i}}\times 100$$
(3)$$Fluid\;uptake\left(\%\right)=\frac{W_{w}-W_{d}}{W_{d}}\times 100$$
where W_i_ is the initial weight of the tablet, W_w_ is the weight of wet tablet and W_d_ is the weight of dry tablet.

The percentage of tablets mass remaining after erosion was calculated from Equation (4).
(4)Remaining mass(%)=100−Erosion(%)

### 3.7. Antioxidant Capacity

The ability to reduce the ABTS^•+^ [2,2-azinobis (3-ethylbenzothiazoline-6-sulfonic acid)] radical cation was evaluated for the ACNs released from C4 tablets. The ABTS^•+^ radical was prepared according to Konan et al. [58]. Aliquots of 50 µL were collected during the dissolution test (in SGF or SIF) and mixed with 950 µL of ABTS^•+^ and the absorbances were recorded after 1 min of reaction at 734 nm. The results were expressed in µM of Trolox equivalence per mL of solution. The decrease of the initial absorbance of the radical (0.700) was evaluated and plotted against the percentage of ACNs released in the referred medium.

### 3.8. Statistical Analysis

The data generated in the experiments were analyzed in Software OriginPro version 8. The experiments were carried out in triplicate and error bars represent the standard error for three measurements.

## 4. Conclusions

The jambolan anthocyanin extract was associated with carboxymethyl starch to obtain optimized formulations for the gastric and intestinal delivery of ACNs. The release of ACNs was controlled by the CMS matrix which is pH sensitive. The dissolution data were well fitted with the Korsmeyer–Peppas model: the release exponent *(n)* revealed variations for the anthocyanin release mechanism mainly due to the pH of the dissolution medium, except when ACNs:CMS was pre-stabilized by protonation in gastric acidity. It was possible to develop dosage-forms able to release 100% of their anthocyanin content from 30 min (immediate-release) to 12 h (sustained release), suggesting that the delivery of anthocyanins in different sites of the gastrointestinal tract could be obtained. The carboxymethyl starch was shown to be an excipient suitable for delivery of ACNs, which may increase their structural stabilization and provide enhanced benefits for human health. This paper presents important information for further studies related to the development of delivery systems for ACNs using carboxymethyl starch as carrier.

## Data Availability

Not applicable.

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
