# Peer review of "Anthocyanins Formulated with Carboxymethyl Starch for Gastric and Intestinal Delivery"

_molecules, 2022, doi:10.3390/molecules27217271_

Round 1

Reviewer 1 Report

This article focuses on the usage of jambolan anthocyanin extract in conjunction with carboxymethyl starch to produce formulations that are ideal for the transport of anthocyanins through the stomach and intestines.

An examination of the release of anthocyanins was performed by the authors, and formulations were devised to allow for the controlled release of these compounds at various places throughout the gastrointestinal tract.

It appears that the current study is the first one to investigate the use of CMS as an excipient for the treatment of ACNs.
The problem that is discussed in the paper is one that is substantial, highly influential, and highly unique. The paper addresses this highly original and novel subject.
It is a pretty original piece of writing that is also quite nicely written.

It has been determined that the experimental and methodological technique is sufficient in terms of seriousness, organisation, and relevance to be published in the state.

This work gives essential information that may be used in further investigations into the design of anthocyanin delivery systems that make use of carboxymethyl starch as the vehicle.
I believe the item should be published in its current form.

Reviewer 2 Report

The authors have presented an expose' on anthocyanin formulated with carboxymethyl starch for gastric and intestinal delivery. While the work as a whole is quite interesting and adequately executed, the authors are encouraged to pay attention to a few points.

The method used for the quantification of anthocyanins in this work is highly questionable. Standard techniques for determination of total anthocyanin content include HPLC and pH differential. The authors should provide a robust justification for their method citing relevant references.

The manner in which the word 'stability' is used throughout this manuscript is inappropriate. Except in cases where there is evidence for improved stability, authors should refrain from insinuating such.

Authors should provide a thorough explanation for the ACN-PVA as well as ACN-mannitol interaction as evidenced by FTIR data.

Authors are advised to seek professional help with English editing. The manuscript is replete with typos and grammatical errors.

Reviewer 3 Report

Comments to the authors

The MS entitled “Anthocyanins Formulated with Carboxymethyl Starch for Gastric and Intestinal Delivery” was thoroughly reviewed. The article is well set, written, discussed, and designed. In my opinion, the authors did a great job in preparing such a nice MS. Apart from these, the authors should add the following.

1. Results should be revised as results2 and discussion

2. Doi of the references should be added.

Reviewer 4 Report

In this paper, the authors investigated the different formulations of ACNs/CMS to produce monolithic tablets for delivery of ACNs into gastrointestinal simulated media. This research is well-designed and performed, some comments are listed as follows:  

1.    In Figure 1C, the SEM of ACNs:CMS complex, what’s the ratio between ACNs and CMC?

2.    In Scheme 2, the hydrogen bond interaction between ACNs and CMC is inaccurate. The hydrogen bond should form between the oxygen atom of ACNs and the hydrogen atom of CMS.

3.    Line 203, what does “in Antman” means?

4.    In Figure 6, F 4.6 and F4.7 also contain CCS and MCC, why is there such a difference in ACNs release compared with 4.8?

5.    What is the sample size in Figures 6 and 7?

6.    The text format should be carefully checked, such as “cm-1” in line 233 should be superscript.
